# Sponsorship Bias in Clinical Trials in the Dental Application of Probiotics: A Meta-Epidemiological Study

**DOI:** 10.3390/nu14163409

**Published:** 2022-08-19

**Authors:** Qin Hu, Aneesha Acharya, Wai Keung Leung, George Pelekos

**Affiliations:** 1Faculty of Dentistry, The University of Hong Kong, Hong Kong SAR 999077, China; 2Dr D. Y. Patil Dental College and Hospital, Dr D. Y. Patil Vidyapeeth, Pune 411018, India

**Keywords:** probiotic, dentistry, periodontal disease, sponsorship bias, meta-epidemiological study

## Abstract

Many experimental and clinical trials have investigated the dental application of probiotics, although the evidence concerning the effects of probiotic supplements is conflicting. We aimed to examine whether sponsorship in trials about dental applications of probiotics is associated with biased estimates of treatment effects. Overall, 13 meta-analyses involving 48 randomized controlled trials (23 with high risk of sponsorship bias, 25 with low risk) with continuous outcomes were included. Effect sizes were calculated from differences in means of first reported continuous outcomes, divided by the pooled standard deviation. For each meta-analysis, the difference in standardized mean differences between high-risk and low-risk trials was estimated by random effects meta-regression. Differences in standardized mean differences (DSMDs) were then calculated via meta-analyses in a random effects meta-analysis model. A combined DSMD of greater than zero indicated that high-risk trials showed more significant treatment effects than low-risk trials. The results show that trials with a high risk of sponsorship bias showed more significant intervention effects than did low-risk trials (combined DSMD, 0.06; 95% confidence interval, 0.3 to 0.9; *p* < 0.001), with low heterogeneity among meta-analyses (*I*^2^ = 0%; between-meta-analyses variance τ2 = 0.00). Based on our study, high-risk clinical trials with continuous outcomes reported more favorable intervention effects than did low-risk trials in general.

## 1. Introduction

Probiotics, in 2002, were defined by the Food and Agriculture Organization of the United Nations and the WHO as “live microorganisms that, when administered in adequate amounts, confer a health benefit on the host” [1].

In recent decades, the benefits of probiotics on general health have been extensively investigated, increasing the consumption of probiotics for the promotion of health and well-being all over the world [2]. Studies reported their potential beneficial impact in treating a variety of conditions such as digestive system diseases [3,4], respiratory diseases [5,6], metabolic diseases [7,8,9], psychological diseases [10,11], and immune system diseases [12,13]. Data in 2020 showed globally that the probiotic market size was valued at USD 34.1 billion and is expected to reach USD 73.9 billion by 2030 [14].

Due to their potential ability for microbiota modification and host regulation, probiotics are also widely studied in dentistry, including periodontal disease [15,16], dental caries [17,18], and halitosis [19]. Consequently, dental probiotics categorized as food supplements are marketed and are being popularized rapidly. Though good scientific support is also required for food and supplements, the commercialization routes and regulatory requirements differ from those for drugs. A total of 350 clinical trials, systematic reviews, and meta-analyses can be identified by searching “probiotics and dental health” in PubMed. Regretfully, even though many studies have been carried out, the overall conclusions are conflicting at best and debatable. The EFP S3 level clinical practice guideline [20] evaluated five placebo-controlled RCTs and concluded that there is no strong evidence regarding the effectiveness of probiotics as adjuncts to subgingival instrumentation. In this guideline, all five included studies were assessed as low-quality with the risk of industrial sponsorship interference.

Evidence suggests that industry sponsorship of research is associated with favorable efficacy results [21]. Possible explanations for the promising results seen in industry-sponsored research [22] include the following:Pharmaceutical companies might fund studies with weaker comparators.Industry may have conducted low-quality research.Higher doses of the drug may be administered to subjects.Manufacturers tend to prevent the publication of studies unfavorable to their products.Sponsored research is more likely to appear in symposiums and use publication platforms with a lack of peer review.

All the above biases adherent to sponsorship are termed as sponsorship bias and might account, in part, for the conflicting results of dental probiotics trials.

In the dental field, up to now, only few publications have evaluated the effect of sponsorship on intervention outcomes [23,24,25]. Two studies [23,24] investigated sponsorship’s impact on implant and restorative dentistry, adopting meta-regression and network meta-analysis. Both reported no significant effect of industry sponsorship on clinical trial outcomes. Recently, one publication used the meta-epidemiological method to evaluate the impact of sponsorship on effect size in trials of oral health interventions and identified significant differences between dental trials [25]. However, no studies concerning probiotics’ application in dentistry were included. Meta-epidemiological studies examine the impact of specific characteristics of clinical studies with estimated treatment effects in a collection of meta-analyses and their component trials [26]. In this study, we carried out meta-epidemiological research of meta-analyses reporting probiotics’ application in dentistry to assess the association of estimates of treatment benefits with sponsorship status.

## 2. Methods

### 2.1. Eligible Meta-Analyses

We sought meta-analyses investigating probiotics’ application for oral disease treatment or maintenance. The inclusion criteria were as follows:Existence of both trials sponsored by industry and trials by non-industrial institutions;At least one qualitative continuous outcome related to oral health;At least three trials included in the meta-analysis.

Exclusion criteria were:Inaccessible trials included;Studies other than randomized control trials included.

Systematic searches were conducted in 3 databases (MEDLINE, EMBASE, and Web of Science), while unpublished papers were searched in OpenGrey. The search strategy for meta-analyses can be found in the Appendix A.

### 2.2. Data Extraction and Assessment of Sponsorship Status

We retrieved each trial publication in eligible meta-analysis studies. For included trials, we extracted the following information (Table 1): author name, publication date, dental fields, first reported continuous outcome in the meta-analysis, the number of patients in each group, and sponsorship status. Regarding the first reported continuous outcome in the meta-analysis, we aimed to extract the change value. In case this was unavailable, we calculated it according to the raw data from the original reports of the trials.

To identify factors affected by or correlated with sponsorship status, we assessed each included trial’s risk of bias according to the Consolidated Standards of Reporting Trials (CONSORT) statement [27]. In accordance with Als-Nielsen et al. [28], the sponsorship bias status was described as “high” or “low.”

The criteria are listed as follows:Trials reporting that they did not receive sponsorship or only received support from universities or other academic institutions were judged as low-risk.Trials reporting that they received sponsorship from industry and academic institutions and declaring the sponsor was not involved in the trial conduct, data management/analysis, or co-authorship were judged as low-risk.Trials reporting that they received sponsorship from industry and academic institutions without declaring the sponsor was not involved in the trial conduct, data management/analysis, or co-authorship were judged as high-risk.Trials reporting that they received sponsorship from the industry and declaring that the sponsor was not involved in the trial conduct, data management/analysis, or co-authorship were judged as low-risk.Trials reporting that they received sponsorship from the industry without declaring that the sponsor was not involved in the trial conduct, data management/analysis, or co-authorship were considered high-risk.

For the whole process of data extraction, two investigators (Q.H. and G.P.) extracted data independently. Any disagreements were resolved after discussion with a third investigator (A.A.).

## 3. Data Synthesis and Analysis

Treatment effects were expressed as standardized mean difference (SMD, based on the adjusted Hedges’ g (95% CI) model; difference in mean outcomes between groups divided by a pooled standard deviation within groups) for the first reported continuous outcomes in each meta-analysis [29]. Due to the differences in the direction of scales, some standardized mean differences were multiplied by −1. An SMD > 0 indicated a beneficial effect in the experimental arm.

The meta-epidemiological analysis relied on previously described methodology [30]. To evaluate the difference in standardized mean differences between trials with low and high sponsorship bias, we used random effects meta-regression to incorporate heterogeneity between trials for each meta-analysis. Then, the differences in standardized mean differences (DSMDs) across meta-analyses were synthesized with a random effects meta-analysis model. The results are reported as the mean difference in standardized mean differences with associated 95% confidence intervals between low-risk and high-risk trials. A DMSD > 0 indicated that, on average, trials with a high risk of sponsorship bias presented larger treatment effects than trials with low risk. The *I*^2^ statistic, Cochran’s Q χ^2^ test, and the between-meta-analyses variance τ^2^ were used to assess heterogeneity across differences in standardized mean differences.

To account for the difference in estimated effects between high-risk and low-risk trials, the domains of the reporting study (including sequence generating, blinding, allocation concealment, incomplete follow-up data, and selective reporting) were assessed using Chi-square tests. Furthermore, we compared the publication bias of high-risk and low-risk trials using the regression-based Egger test [31] with a visual inspection of the funnel plot.

All statistical analyses were performed using Stata software version 17.0 (StataCorp LP, College Station, TX, USA). Statistical significance was considered at a two-sided *p* < 0.05.

## 4. Results

### 4.1. Eligible Meta-Analyses for Continuous Outcomes

The initial search identified 285 citations, of which 83 were removed due to duplication, while 15 meta-analyses fulfilled the inclusion criteria after reviewing the title, abstract, and full text. However, two meta-analyses were excluded since they included inaccessible trial publications. The final study database contained 13 meta-analyses [32,33,34,35,36,37,38,39,40,41,42,43,44] with 48 trials, of which 25 trials were judged as having low risk of sponsorship bias, while 23 were considered to have a high risk. Figure 1 shows the flowchart of the screening process. The description of each meta-analysis’s characteristics and the quality assessment of the trials can be found in Table 1 and Appendix A. These meta-analyses were in the fields of periodontal diseases, peri-implant disease, caries, oral mucosal diseases, halitosis, and oral health maintenance.

Among those 48 trials, most (12 trials) adopted probiotic products from BioGaia^®^ (Sweden), six obtained products from Sunstar^®^ (Switzerland), and the rest of the trials obtained probiotics from other companies. *Lactobacillus* and *Bifidobacterium* were the two most common genera used.

**Table 1 nutrients-14-03409-t001:** Characteristics of included meta-analyses.

	**Condition**	**Experimental Intervention**	**Control** **Intervention**	**Outcome**	**Trials in High Risk, N**	**Trials in Low Risk, N**	**All Trials, N**
Cheng 2020 [32]	Recurrent aphthous stomatitis	Treatment with probiotics, either alone or combined with other drugs	Treatment with placebo or other drugs alone	Visual Analogue Pain Scale	2	1	3
Donos 2020 [33]	Periodontitis	Probiotics	Placebo	Periodontal probing depth reduction	1	4	5
Gao 2020 [34]	Peri-implant diseases	*Lactobacillus* agent	Placebo agent or blank control	Periodontal probing depth reduction	3	1	4
Gheisary 2022 [35]	Periodontal diseases/health	Probiotics in any form	Without probiotics, with a placebo, or with antibiotics	Plaque index	6	10	16
Hao 2021 [36]	Caries	Products containing *Bifidobacterium*	Products without *Bifidobacterium*	*Streptococcus mutans* counts	3	1	4
Hu 2021 [37]	Periodontitis	Scaling and root planning + probiotics	Scaling and root planning	Periodontal probing depth reduction	4	4	8
Ikram 2018 [38]	Periodontitis	Scaling and root planning + probiotics	Scaling and root planning alone or with a placebo	Periodontal probing depth reduction	1	2	3
Martin-Cabezas 2016 [39]	Periodontitis	Scaling and root planning + probiotics	Scaling and root planning alone or with a placebo	Periodontal probing depth reduction	1	2	3
Mishra 2021 [40]	Periodontitis	Scaling and root planning + probiotics	Scaling and root planning + placebo	Periodontal probing depth reduction	1	2	3
Nadelman 2018 [41]	Oral health establishment	Consumption of probiotic-containing dairy products	Consumption of dairy products without probiotics, other interventions/products, or no intervention	*Streptococcus mutans* counts	5	4	9
Sang-Ngoen 2021 [42]	Caries	Orally administered probiotics	Placebo or no orally administered probiotics	*Aggregatibacter actinomycetemcomitans* counts	1	2	3
Yoo 2019 [43]	Halitosis	Probiotics	Placebo	Volatile sulfur compounds and organoleptic scores	1	2	3
Zhao 2020 [44]	Peri-implant mucositis	Mechanical debridement + probiotics	Mechanical debridement + placebo or alone	Periodontal probing depth reduction	2	2	4

### 4.2. Estimates of Treatment Effect Differences between High-Risk and Low-Risk Trials

Overall, 23 trials with and 25 trials without sponsorship risk were collected from the included 13 meta-analyses. In the meta-regression, the direction of effects was standardized so that DSMD > 0 indicates a larger intervention effect estimate in trials with sponsorship bias versus without sponsorship bias. On average, treatment effects in trials with a high risk of sponsorship bias were more significant than those in low-risk trials (combined DSMD, 0.6; 95% confidence interval, 0.3 to 0.9; *p* < 0.01; Figure 2). The overall estimate effect of studies with sponsorship bias is 0.6 times greater than that of those without sponsorship bias. The estimated differences in standardized mean differences were positive for ten meta-analyses and negative for three meta-analyses. Heterogeneity across individual meta-analyses was low (*I*^2^ = 0%; between-meta-analysis variance τ^2^ = 0.00).

### 4.3. Reporting Quality Comparison

The Cochrane Collaboration’s tool for assessing the risk of bias in randomized trials (RoB) is the most commonly used RCT quality evaluation tool. Considering previous publications which raised the hypothesis that poor quality might account for the conflicting results of clinical trials [22], we compared reporting qualities between high sponsorship bias risk trials and low sponsorship bias risk trials using the RoB tool (Table 2) to explore possible explanations for the difference in effect sizes. As expected, trials with a low risk of sponsorship bias presented high quality in reporting in general. A higher percentage (68.2%) of low-risk trials explicitly reported randomization, with a lower percentage (31.8%) of high-risk trials doing so (*p* = 0.01). In addition, a higher proportion (65.7%) of trials at low risk performed blinding of outcome assessment, while a lower proportion (52.2%) of high-risk trials performed blinding of outcome assessment (*p* = 0.02). Though we did not observe significant statistical differences for items such allocation concealment, blinding of outcome assessment, and selective reporting, the proportion of low-risk studies reporting a better quality was higher than that of high-risk studies.

### 4.4. Risk of Publication Bias

To investigate whether the industry might interfere with the publication of unfavorable results for their product, we adopted the Egger test, which revealed no presence of publication bias (*p* > 0.05) in both groups, which was confirmed by the funnel plot analysis (Figure 3). No statistically significant difference was noted between sponsorship categories. However, compared with the lower limit of the 95% confidence interval of the low-risk group (3.34, 95% CI from −3.54 to 10.22), the lower limit of the 95% confidence interval of the high-risk group was closer to zero (2.871, 95% CI from −1.17 to 6.91). There was a slight tendency for high-risk trial treatment effects to be asymmetrically distributed, which might indicate a publication bias.

Another possible source of bias could be trial registration. Here, we found that the majority of the trials did not declare registration, and this factor could not be assessed.

## 5. Discussion

This is a meta-epidemiological study conducted in the field of dentistry to demonstrate that industry-sponsored trials may overestimate treatment effect size. This study included 13 meta-analyses of 48 randomized controlled trials covering different specialties, including periodontology, endodontics, implantology, preventive dentistry, and oral mucosal diseases. Overlapping trials existed between different meta-analyses. The meta-epidemiological method allows trials to be contained in different meta-analyses [45]. Overlap between meta-analyses is an inevitable challenge in meta-epidemiological studies, especially for those with a very specific research scope. Different statisticians may adopt different decisions with the repeated studies for different reasons. On the one hand, overlapping studies were considered to need to be removed to avoid inflating the results; on the other hand, researchers have suggested that overlap might be enriching for overviews if statistical dependencies are properly addressed [46]. For our study, we chose not to remove them from the meta-regression model for two reasons: Firstly, for this topic, we only obtained 13 meta-analyses by exhausting the database. If we removed the overlapping meta-analyses, it would largely lead to data loss. Secondly, in the rest of the study, we performed quality assessment and publication assessment to detect their relationship with sponsorship bias, which could be taken as a compensating or sensitivity analysis. They are based on non-overlapping trials instead of meta-analysis.

The results of our study are in accordance with or can explain the results from the meta-regression. The results showed that trials with high sponsorship risk were more likely to report larger estimated treatment effects than low-risk trials, and the heterogeneity between studies was relatively low. Furthermore, the high-risk trials were of lower quality in methodology and tended to have greater publication bias compared with low-risk trials, which may be partly accounted for by sponsorship bias.

Some trials included in our study which did not clearly indicate the sponsorship status were classified as trials with a high risk of sponsorship bias. Some might argue that our classification was not stringent enough, and an “unclear” group could be included in this study, but we intended to uncover any possible bias associated with sponsorship status in the analysis. Information about the sponsorship status should be clearly indicated in future studies. Apart from the basic information required by CONSORT, explicit information about the sponsorship status should be given when reporting trials. If the study was sponsored, the sources of sponsorship; the type of resources received; and whether the sponsor joined the design, data analysis, and publication of the study should be listed. Systematic reviews or meta-analyses should perform a sensitivity analysis when including trials with high risk to avoid the effects of sponsorship bias.

Secondly, although the heterogeneity between the studies was not significant for the outcomes analyzed, different strains of probiotics were investigated in each trial. Therefore, our study cannot determine the influence of different probiotic strains on the outcomes. This might explain why the overall results of different systematic reviews and meta-analyses, including studies with different probiotics, conflicted with each other. To obtain more accurate information, future systematic reviews and meta-analyses can review studies with homogeneity in the research object (probiotics of the same species). Overall, our results regarding the application of probiotics in dentistry displayed a significant difference in the effect size between trials with and without sponsorship bias, as seen in studies of pharmaceuticals [22], devices [21], and psychotherapy [47].

## 6. Conclusions

This study revealed that the effect size estimates of industry-sponsored trials were 0.6 times greater than those of trials without industrial sponsorships, indicating a sponsorship bias. Therefore, we suggest that the design, analysis, interpretation, and reporting of clinical trials investigating probiotics should follow the Consolidated Standards of Reporting Trials (CONSORT) to ensure study quality in trials of probiotics within dentistry.

## Figures and Tables

**Figure 1 nutrients-14-03409-f001:**
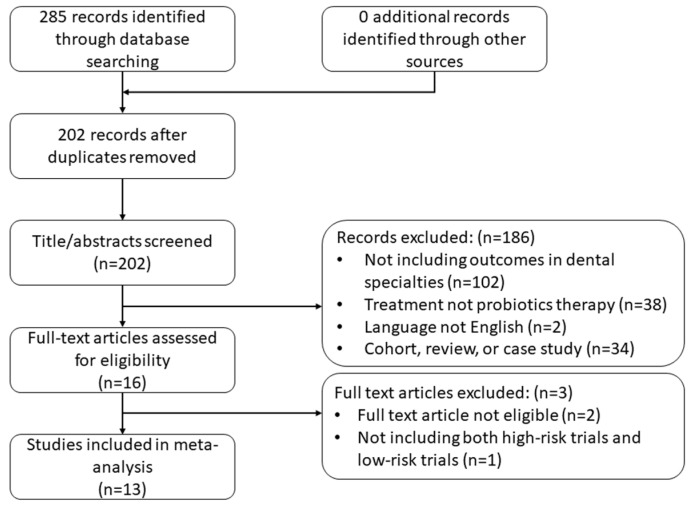
PRISMA flow diagram.

**Figure 2 nutrients-14-03409-f002:**
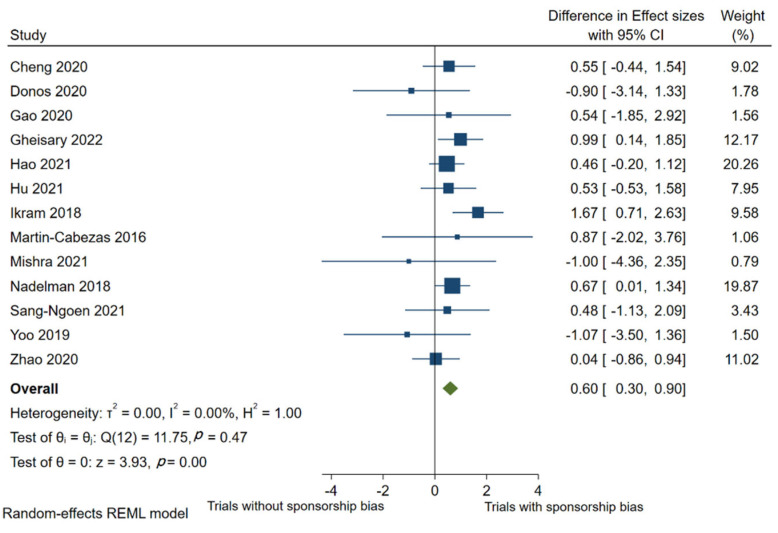
Difference in effect sizes between 23 trials with and 25 trials without sponsorship risk from 13 meta-analyses using meta-regression. A positive difference in effect sizes indicates that trials with sponsorship risk show more beneficial treatment effects. The effect size estimates of industry-sponsored trials were 0.6 times greater than those without industrial sponsorship (DSMD = 0.6, *p* < 0.001) [32,33,34,35,36,37,38,39,40,41,42,43,44].

**Figure 3 nutrients-14-03409-f003:**
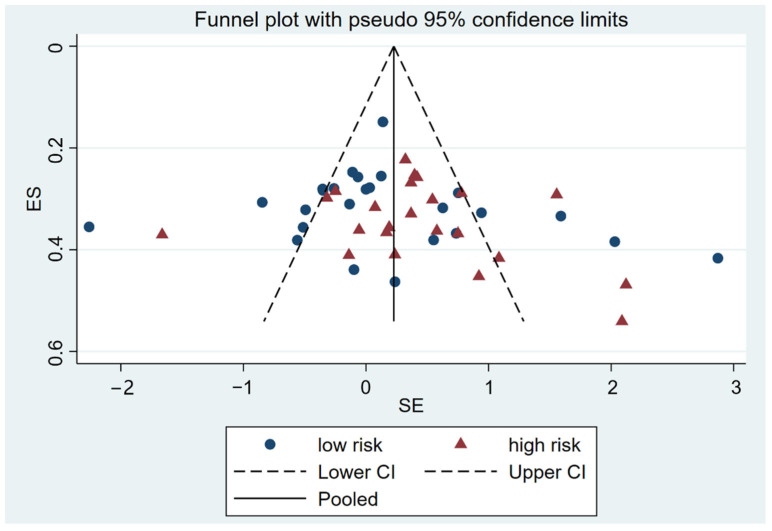
Meta-funnel plotted in the first reported continuous results in the meta-analysis for each trial against their standard errors. Different colors indicate different sponsor categories. The outer dashed lines indicate the triangular region within which 95% of studies are expected to lie in the absence of both biases and heterogeneity.

**Table 2 nutrients-14-03409-t002:** Comparison of qualities between high-risk trials and low-risk trials.

	Risk of Sponsorship Bias		*p*-Value
	Low	High	
Randomization			0.01 *
Adequate	18 (69.2%)	8 (31.8%)	
Inadequate/unclear	7 (30.8%)	15 (68.2%)	
Allocation concealment			0.414
Adequate	18 (72%)	14 (60.9%)	
Inadequate/unclear	7 (28%)	9 (30.1%)	
Blinding of participants			0.02 *
Adequate	23 (65.7%)	12 (52.2%)	
Inadequate/unclear	2 (15.4%)	11 (47.8%)	
Blinding of outcome assessment			
Adequate	21 (92%)	21 (91.3%)	0.445
Inadequate/unclear	4 (8%)	2 (8.7%)	
Incomplete outcome data			0.331
Adequate	19 (76%)	20 (87%)	
Inadequate/unclear	6 (24%)	3 (13%)	
Selective reporting			0.606
Adequate	17 (68%)	14 (60.9%)	
Inadequate/unclear	8 (32%)	9 (39.1%)	

* *p* value < 0.05.

## Data Availability

Not applicable.

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
