# Peer review of "Sponsorship Bias in Clinical Trials in the Dental Application of Probiotics: A Meta-Epidemiological Study"

_nutrients, 2022, doi:10.3390/nu14163409_

Round 1
Reviewer 1 Report
Very well written and argued overall. Meta-analyses on the level of multiple systematic reviews are still relatively new in the area of dentistry. There is one part in the meta-analysis of the treatment effect differences between the high and low bias studies (Figure 2) that may need some clarification from the statistician. The overall calculated effect difference includes systematic reviews like Donos 2020 and Ikram 2018 that in turn include some of the same original studies e.g. Ince et al. 2015 or similarly the Morales et al. 2016 RCT in both the Donos 2020 and Mishra et al 2021 systematic reviews . Have you and how adjusted, accommodated for this, i.e. some original RCTs been represented in more than one systematic review, and if not can this been affecting/biasing your overall results?
Reviewer 2 Report
Very interesting study.
It could be helpful to have a vertical 0 reference line in Figure 2.
Line 51-58 may need some help with formatting.
Could proper clinical trial registration (registration prior to patient recruitment) be included in Table 2. This could provide information on whether endpoint switching is more likely in industry funded trials.
I personally would like to see p-values in Figure 2.
Figure description is a bit confusing -the description refers to 23 trials with and 25 trials without sponsorship and one is tempted to look for 48 trials in the figure. But what is in the figure is 13 meta-analyses - this could be made clearer by improving the description of the Figure.
I would like to have some information on how these 13 meta-analysis - how many of these were funded by industry? How many of the 48 trials were included in all 13 meta-analyses?
